# The Design, Modeling and Experimental Investigation of a Micro-G Microoptoelectromechanical Accelerometer with an Optical Tunneling Measuring Transducer

**DOI:** 10.3390/s24030765

**Published:** 2024-01-24

**Authors:** Evgenii Barbin, Tamara Nesterenko, Aleksej Koleda, Evgeniy Shesterikov, Ivan Kulinich, Andrey Kokolov, Anton Perin

**Affiliations:** 1Research Institute of Microelectronic Systems, The Tomsk State University of Control Systems and Radioelectronics, 634050 Tomsk, Russia; evgenii.s.barbin@tusur.ru (E.B.); ntg@tpu.ru (T.N.); nanocenter@tusur.ru (E.S.); kulinich@tusur.ru (I.K.); andrei.a.kokolov@tusur.ru (A.K.); anton.s.perin@tusur.ru (A.P.); 2Laboratory of Radiophotonics, V.E. Zuev Institute of Atmospheric Optics SB RAS, 634055 Tomsk, Russia; 3Division for Electronic Engineering, National Research Tomsk Polytechnic University, 634050 Tomsk, Russia

**Keywords:** microoptoelectromechanical accelerometer, tunneling effect, evanescent coupling, proof mass, waveguides, microoptoelectromechanical sensing element, optical measuring transducer, sensitivity threshold, positioning system

## Abstract

This treatise studies a microoptoelectromechanical accelerometer (MOEMA) with an optical measuring transducer built according to the optical tunneling principle (evanescent coupling). The work discusses the design of the accelerometer’s microelectromechanical sensing element (MSE) and states the requirements for the design to achieve a sensitivity threshold of 1 µg m/s^2^ at a calculated eigenvalue of the MSE. The studies cover the selection of the dimensions, mass, eigenfrequency and corresponding stiffness of the spring suspension, gravity-induced cross-displacements. The authors propose and experimentally test an optical transducer positioning system represented by a capacitive actuator. This approach allows avoiding the restrictions in the fabrication of the transducer conditioned by the extremely high aspect ratio of deep silicon etching (more than 100). The designed MOEMA is tested on three manufactured prototypes. The experiments show that the sensitivity threshold of the accelerometers is 2 µg. For the dynamic range from minus 0.01 g to plus 0.01 g, the average nonlinearity of the accelerometers’ characteristics ranges from 0.7% to 1.62%. For the maximum dynamic range from minus 0.015 g to plus 0.05 g, the nonlinearity ranges from 2.34% to 2.9%, having the maximum deviation at the edges of the regions. The power gain of the three prototypes of accelerometers varies from 12.321 mW/g to 26.472 mW/g. The results provide broad prospects for the application of the proposed solutions in integrated inertial devices.

## 1. Introduction

Microelectromechanical accelerometers are widely used in numerous engineering spheres. Most accelerometers include a proof mass mounted on a fixed substrate on a spring suspension. The proof mass displaces under the inertia forces induced by acceleration. The proof mass displacement can be converted into a measurable signal using various methods, such as piezoresistive, piezoelectric, thermal, electrostatic, and optical techniques, electron tunneling, etc. [1,2,3,4,5,6,7]. Microoptoelectromechanical accelerometers have become increasingly popular in recent times because they incorporate the advantages of optical measurements and microelectromechanical systems. They are immune to electromagnetic noise, and they have high corrosion resistance, high sensitivity and a wide range of applications: high-precision inertial navigation, vibration monitoring, equipment monitoring, structure and vehicle monitoring, seismic exploration, and the oil industry [8,9,10,11,12,13,14].

In MOEM accelerometers, the displacement of the proof mass alters the characteristics of the output light flux. The designs of MOEM accelerometers implement various optical methods to measure the displacement of the proof mass, including grating interferometry, Fabry–Pérot cavity, fiber Bragg grating (FBG), photonic crystal nano-cavity, and optical tunneling [15,16,17,18,19,20,21,22]. Particularly promising are MOEMS technologies that integrate lasers, photodiodes and passive elements in a single technological cycle [23].

For instance, in [24], the authors have developed an optimal mechanical design for a MOEM accelerometer that implemented a diffraction grating-based interferometer to measure the displacement of the proof mass under acceleration. The accelerometer of such a design reached a sensitivity to proof mass displacement of about 169 µm/g and a sensitivity to acceleration of about 60 V/g. The experiments have shown that the intrinsic noise of the MOEM accelerometer did not exceed 15 ng/√Hz at 1 Hz.

Mireles et al. [25] described the design, manufacturing and testing of a MOEM accelerometer consisting of a structure mounted in a fixed substrate on a spring suspension and a sensing system based on a Fabry–Pérot interferometer. The accelerometer implemented a custom-designed spring suspension for the proof mass. The paper presented the main parameters of the system, including the frequency characteristics, mass, stiffness of the elastic elements and damping coefficient. The accelerometer reached a resonance frequency of 1274 Hz and a damping coefficient of 0.0173 at measured accelerations from 0.2 to 7 g. The experimental results showed that the designed and manufactured MOEM accelerometer is a promising instrument for vibrational monitoring under highly intense environmental electromagnetic noise.

The majority of MOEM accelerometers are uniaxial, because the optical conversion of the displacement into an electrical signal should allow for distinguishing the direction of the proof mass displacement, which is not always possible for the aforementioned methods. However, Abozyd et al. [26] presented a triaxial MOEM accelerometer in which the proof mass was suspended on four elastic elements that allowed it to displace in three dimensions. The proof mass displacement was altering the light flux traveling to the four optical detectors. The presented optical accelerometer achieved a sensitivity of 0.156 mA/g and a resolution of 56.2 µg, having dimensions of 1.5 mm × 4 mm × 4 mm. However, the proposed triaxial suspension system can be implemented in a limited range of optical methods for proof mass displacement measurement.

The biaxial MOEM accelerometer described in [27] consisted of the proof mass mounted on four L-shaped elastic elements. Two Fabry–Pérot cavities were formed between the transverse cross-sections of the proof mass and the ends of two optical fibers cleaved in perpendicular directions (X, Y). The accelerometer had a measurement range of ±1 g and displacement sensitivities of 3.23 nm/g and 3.19 nm/g for applied accelerations along the X and Y axes, respectively. The resolutions for the two directions, X and Y, were 309 and 313 μg; the eigenfrequencies were 1382.5 Hz and 1398.6 Hz along the X and Y axes, respectively. The maximum sensitivities along the transverse axes were less than 0.19% for each pair of X, Y and Z axes.

Sun et al. [28] presented the design of a monolithic MOEM accelerometer with an optical sensor based on the Michelson interferometer. The accelerometer had a push–pull structure, which eliminates the coupling crosstalk caused by paraxial acceleration. The resulting characteristics were as follows: the mechanical sensitivity of the accelerometer was 3.638 nm/g, the eigenfrequency was 1742.2 Hz, the linear measurement range was ±500 g, and the sensing element dimensions were 960 μm × 600 μm.

Chen at al. [29] proposed a super-sensitive uniaxial accelerometer with an optical measuring transducer based on the Talbot diffraction effect on dual-layer gratings. The eigenfrequency of the accelerometer was 1878.9 Hz, the mechanical sensitivity was 140 nm/g, the sensitivity to acceleration was 0.74 V/g, the displacement stability was 75 μg and the acceleration resolution was 2.0 mg.

The combination of MEMS with integrated photonics requires special engineering approaches and the integration of technological fabrication processes. Currently, one of widely applied methods is the formation of the fiber Bragg grating (FBG) immediately in the fiber. However, the integration of the FBG in MEMS devices and the expansion of their application require more sophisticated combined manufacturing and assembly of the optical and mechanical parts of the accelerometer. Moreover, the resonance methods the fiber Bragg grating is based on are the most sensitive to temperature and require much more sophisticated electronics and precise adjustment of the resonance that is affected by the temperature.

The accelerometers operating on the basis of light intensity modulation are economically feasible and imply a simpler manufacturing process as compared to those incorporating the fiber Bragg grating. A MOEM accelerometer designed and manufactured by Gholamzadeh [30] was based on light intensity modulation. The accelerometer had the following characteristics: the resonance frequency was 560 Hz, mechanical sensitivity was 600 nm/g, optical sensitivity was 16%/nm, general sensitivity was 9.6%/g, dimensions were 2 mm × 2 mm, measurement range was 3 g, and mechanical sensitivity along the transverse axis was 58 nm/g. Soltanian et al. [31] suggested a highly sensitive differential optical MEMS accelerometer based on light intensity modulation involving double optical outputs and a two-dimensional photonic-crystal power splitter. The accelerometer had a mechanical sensitivity of 0.0750 nm/g and a linear measurement range of ±200 g. The mechanical resonance frequency of the accelerometer was 17.7 kHz, and the optical sensitivity was 4.38%/nm.

The directional coupler is the basic functional element of the MOEM accelerometer’s optical transducer that has the highest practical potential. The designs of MOEM accelerometers with optical displacement sensors based on the tunneling effect are highly promising in terms of the achievable compactness and high sensitivity [32,33,34]. To increase the sensitivity, optical resonators are used; however, they are highly sensitive to temperature. In various sources, the authors present the characteristics of MOEM accelerometers in different units at their reasonable discretion. Moreover, the outputs of accelerometers depend on the type of the optical measuring transducer, the value of inertial mass, transmission coefficients of photodiodes and other optical and electrical components. Table 1 summarizes the literature review.

The worldwide literature lacks studies describing the application of the optical tunneling effect for measuring acceleration, which places the research in this sphere high on the agenda. This work presents the theory, modeling, and experimental demonstration of a MOEM accelerometer based on the optical tunneling effect. Such an accelerometer allows for achieving the high sensitivity and resolution of acceleration measurement thanks to the acceleration-sensitive chip of the MOEM accelerometer. The paper presents the design and optimization of the mechanical and optical parts of the proposed accelerometer. The analysis and numerical modeling involved the Finite Difference Time Domain and Finite Element Methods.

## 2. Designing Micromechanical Sensing Element of Accelerometer

### 2.1. Functional Scheme

The MOEM accelerometer consisted of a micromechanical sensing element (MSE) reacting to acceleration and an optical measuring transducer (OMT). The MSE converted the applied acceleration into displacement; the OMT measured the MSE’s displacement under acceleration. The combined characteristics of both parts determined the general characteristics of the MOEM accelerometer.

Figure 1 depicts the functional scheme of the MOEM accelerometer. The MSE contained proof mass 1 mounted on elastic suspension 2 in fixed substrate 3. The OMT functionally consisted of moving waveguide 4 fabricated on the MSE and fixed waveguide 5 fabricated on substrate 3. The fixed and moving waveguides lay in a single plane and were separated by an air gap. The MSE with the moving waveguide had the degree of freedom along the Y axis. The optical radiation from laser diode 6 was fed through the input port to the fixed waveguide and exited through the output port to photodiode 7. For such a design of the waveguides and working wavelength, the distance between the waveguides and coupling length determined the amount of power transmitted from the input port to the output one. Under acceleration a_y_, the proof mass displaced along the Y axis; the gap between the moving and fixed waveguides changed (Figure 1), which altered the output optical power *P_opt_* that was proportional to the measured acceleration *a_y_* [34].

According to the accelerometer fabrication process (Section 3), the initial gap formed between the waveguides exceeded its working value. Therefore, the gap between the waveguides was adjusted via an optical transducer positioning system (OTPS) that was presented by comb electrode structure 8. The moving electrodes of the OTPS were coupled with the proof mass. The fixed electrodes were mounted on the fixed substrate. The voltage applied to the comb electrode structure induced the electrostatic force that displaced the MSE, which changed the gap between the moving and fixed waveguides. The OTPS can be improved by implementing several stages with different numbers of electrodes. The positioning system is described in detail in Section 2.2.3 and paper [35]. Thus, the micromechanical sensing element of the accelerometer consisted of the proof mass with the moving waveguide of the optical measuring transducer and the moving electrodes of the OTPS mounted on it.

### 2.2. Mechanical Characteristics of the MSE

#### 2.2.1. Dimensions, Mass, and Eigenfrequencies of the MSE

To design the micromechanical sensing element, we analyzed and prognosticated the impact of the geometric parameters of the structure, limitations of the technological microfabrication process and operational conditions of the device on the resulting characteristics of the transducer. We have identified the following main parameters of the accelerometer’s MSE: dimensions, mass, eigenfrequencies along the Y and Z axes and corresponding stiffness of the spring suspension formed from several elastic elements. Such parameters determined the mechanical sensitivity (transformation coefficient) of the MSE, i.e., its displacement under acceleration. Such displacement was then transformed by the optical system into the output signal. Therefore, high sensitivity, low cross-talk and a certain bandwidth were achieved by adjusting the geometric dimensions of the accelerometer’s MSE.

The accelerometer’s resolution was quantitatively estimated using the Noise Equivalent Acceleration, which consisted of the mechanical thermal noise, laser noise and electrical scheme noise present during the measurement of the accelerometer’s MSE displacement. The value of the Noise Equivalent Acceleration determined the threshold sensitivity of the accelerometer, i.e., the minimal measurable signal.

The acceleration that is equivalent to the mechanical thermal noise of the accelerometer’s MSE can be determined as follows [14,32]:(1)$$a_{BR}=\sqrt{\frac{4\cdot k_{b}\times T\times \omega }{M\times Q}}=\sqrt{\frac{4\times k_{b}\times T}{Q}\times \sqrt{\frac{K}{M^{3}}}},$$
where: *k_b_* is the Boltzmann constant equal to 1.38 × 10^−23^ J/K; *T* is absolute temperature, K; *Q* is the mechanical Q-factor of the accelerometer; *ω* is the eigenfrequency of the MSE on the spring suspension, s^−1^; *M* is the mass of the MSE, kg; and *K* is the stiffness of the MSE’s spring suspension, N/m. Brownian noise contributes the most to the Noise Equivalent Acceleration and can be reduced by increasing the mass and the Q-factor of the accelerometer’s MSE and by decreasing the eigenfrequency. To decrease the transient time, during acceleration measurement, the Q-factor should be minimal. In our case, the Q-factor was equal to 20.

Following Equation (1), the measurement of small accelerations requires the large mass, small eigenfrequency and high Q-factor of the MSE. However, to reduce the transient time, accelerometers usually implement a low Q-factor design, which demands a larger MSE mass to provide high resolution. At the same time, the improved bandwidth of the accelerometer requires a higher eigenfrequency of the MSE, which causes very small displacements that should be measured using the OMT with high resolution. When the frequency of the measured acceleration is much smaller than the resonant frequency of the MSE, its displacement ∆*_i_* is proportional to the measured acceleration:(2)$$\Delta_{i}=\frac{a}{{\left(f_{i}\times 2\times \pi \right)}^{2}},$$
where *f_i_* (Hz) is the eigenfrequency of the MSE along the *i*-th axis (Y or Z).

Figure 2 presents the displacements of the MSE along the sensitivity Y axis under an acceleration of 1 µg depending on the eigenfrequency *f_y_* of the MSE ranging from 10 to 1000 Hz.

Evidently, from Figure 2, an increased eigenfrequency reduced the displacements of the MSE below several picometers, which appreciably limited their detection. Therefore, further investigation and development of the accelerometer involved five configurations of MSEs with eigenfrequencies *f_y_* = (10, 20, 50, 100, 200) Hz.

In general, a massive MSE is hard to obtain; in addition, this increases its dimensions. The decreased stiffness of the spring suspension is a less demanding method to reduce the resonant frequency. Since the MSE’s eigenfrequency depends on the spring suspension stiffness, let us determine it depending on the MSE’s mass at selected frequencies. Figure 3 presents the dependencies of the spring suspension stiffness on the MSE’s mass at different values of eigenfrequency.

Evidently, the decrease in the eigenfrequency at the constant MSE’s mass decreased the spring suspension stiffness. To determine the MSE’s dimensions, we have considered its dimensions and dependencies on mass presented in Figure 3.

Two types of MSEs with the same mass can be manufactured. First, MSEs with the same thickness of the spring suspension and proof mass along its whole surface (Figure 4a). Second, MSEs with a smaller surface area and an additional mass to preserve the required mass (Figure 4b), i.e., the thickness of the proof mass and spring suspension are different.

For the Type 1 MSE, the dimension *ν* determining its area was calculated as
(3)$$v=\sqrt{\frac{M}{\rho \times h}},$$
where *M* is the MSE’s mass; *ρ* is the density of silicon; and *h* is the transducer’s height (35 μm).

Since the additional mass was fabricated via liquid etching of silicon, its side slope was 54.7°. To determine the MSE’s dimensions with the additional mass, the SolidWorks CAD system was used, having assumed the masses of both MSE types to be equal.

Following the plots (Figure 5), the increased mass has enlarged the difference between the areas occupied by the MSEs of both types. For a mass of *M* = 0.01 × 10^−6^ kg, the area occupied by the MSE with additional mass was 2.53 times less than that of the MSE without it. For a mass of *M* = 1 × 10^−6^ kg, the area of the Type 2 MSE was 10.66 times less than that of the Type 1 MSE.

Therefore, the fabrication of the Type 1 MSE yields a smaller number of transducers from a single silicon-on-insulator (SOI) wafer, as compared to the Type 2 MSE with the additional mass (the masses of both MSEs being equal). In addition, for a fixed height of the Type 1 MSE, the increase in its linear dimensions decreases its proof mass stiffness, which causes its bending under gravity and inertial forces. Such bending will also induce an inflection of the moving waveguide, which should be taken into account as well. Hence, to decrease the dimensions of the accelerometer’s MSE, prevent its bending due to the increased size and fabricate more transducers from a single SOI wafer, the design of the MSE with the additional mass should be used.

Figure 6 depicts the dependencies of the thermal mechanical noise *a_BR_* on the mass and stiffness, which correspond to the set frequency. A tenfold increase in the mass (regardless the eigenfrequency) decreased the noise 3.16 times while increasing the occupied area 6.53 times. This affects the number of sensors fabricated from a single SOI wafer.

The value of the thermal mechanical noise should be at least two times smaller than the accelerometer’s sensitivity threshold (1 μg). Therefore, the noise should not exceed 4.905 × 10^−6^ m/s^2^. Following all the above, the MSE’s mass at eigenfrequencies from 10 to 200 Hz should exceed 2.3 × 10^−6^ kg to measure a minimal acceleration of 1 μg.

#### 2.2.2. Spring Suspensions of the MSE

The mechanical elements of the accelerometer also included the MSE’s spring suspension formed from 4 elastic elements, the stiffness of which depended on the geometric parameters denoted in Figure 7. For the spring suspension, eigenfrequencies were determined both along the sensitivity Y axis (*f_y_*) and along the transverse Z axis (*f_z_*).

At the first stage, the length of the elastic elements was selected in such a way that the eigenfrequency *f_y_* of the proof mass corresponded to 10, 20, 50, 100 and 200 Hz. The dimensions *b* and *b*_1_ were fixed and amounted to 4 and 20 μm, respectively; the height of the elastic elements was 35 μm. For each mass and frequency *f_y_*, the corresponding frequencies along the transverse axis *fz* and displacement Δ*z* were calculated. The results are presented in Table 2 and Table 3.

The results demonstrate that the increase in the mass and frequency elongated the elastic elements in a wide range from 353.7 to 23,510 μm. The MSE’s displacement along the transverse Z axis ranged from 82.6 nm to 33.24 μm. Since the moving and fixed waveguides of the OMT should lie in a single plane, large gravity-induced displacements (larger than the waveguide’s height of 350 nm) will render the OMT inoperable. Thus, the frequencies along the sensitivity axis *f_y_* should exceed 100 Hz. At the same time, the reasonable length of the elastic elements should be proportional to the dimensions of the proof mass, so the eigenfrequency *f_y_* of 200 Hz was selected for the further design and experimental studies.

In the following, we studied the effect of the geometric parameters *b*, *b*_1_ and *G* of the elastic elements (Figure 7) on the displacement along the transverse Z axis for a maximum mass of 1.00 × 10^−5^ kg and eigenfrequency *f_y_* of 200 Hz. At different values of *b*, we selected an eigenfrequency *f_y_* of 200 Hz through the alteration of the length of the elastic element *G*. Then, we determined *d_z_* by varying *b*_1_ and matching *G* with the eigenfrequency *f_y_*. The results are presented in Figure 8.

Following Figure 8, the optimal value of *b*1 should exceed 20 μm, while *b* should not exceed 5 μm to minimize the displacement along the Z axis. Then, *G* will be below 500 μm.

#### 2.2.3. Optical Transducer Positioning System

An OTPS is a conventional capacitive actuator that is widely used in the feedback systems of MEMS accelerometers and gyroscopes [36]. During the adjustment of the working gap, the MSE was set into the zero position and compensated its gravity-induced displacement (Section 5.2).

The equation of the accelerometer’s movement is as follows:(4)$$\ddot{y}+\frac{\omega_{y}}{Q}\dot{y}+\omega_{y}^{2}=\frac{1}{M}F_{e}+a_{y},$$
where *M*, *Q*, *K*, and *ω_y_* are the mass, Q-factor, stiffness of the spring suspension and the eigenfrequency of the micromechanical structure ($\omega_{y}=2\pi f_{y}=\sqrt{\frac{K}{M}}$); $F_{e}=\frac{1}{2}\frac{\partial C}{\partial x}U^{2}$ is the electrostatic force generated by the comb electrode structures, which are used to set the working gap; *a_y_* is the measured acceleration; *C* is the capacity of the electrodes; and *U* is the electrode voltage.

Following the optimization of the geometric parameters of the elastic elements, selection of the eigenfrequency *f_y_* of 200 Hz and determination of the minimal mass from the Brownian noise (2.3 × 10^−6^ kg), the accelerometer’s MSE was designed (Figure 9).

In Figure 9, EE are the elastic elements of the MSE’s suspension system; *C*1*_l_* and *C*1*_r_* are the comb electrode structures used to set the initial working gap in the positive direction along the sensitivity Y axis; *C*2*_l_* and *C*2*_r_*, *C*2*_lS_* and *C*2*_rS_*, *C*3*_l_* and *C*3*_r_*, *C*3*_lS_* and *C*3*_rS_* are the comb electrode structures used to simulate the acceleration in the positive and negative directions along the sensitivity Y axis; and the comb electrode structures *C*1*_l_* and *C*1*_r_*, *C*1*_lS_* and *C*1*_rS_* are used to compensate for the MSE’s gravity-induced displacement along transverse axis Z.

To set the working gap (displacement along the sensitivity Y axis), the comb structure was used on one side of the transducer, to which necessary voltage was applied. To compensate for the displacement along the transverse Z axis, two pairs of comb electrode structures located on two sides of the MSE were used. In this case, the electrostatic force worked against gravity and compensated for the arising displacement of the MSE.

## 3. Optical Measuring Transducers

Figure 10 presents the external view of the directional coupler with the denoted ports: 1—Input, 2—Through, 3—Drop, and 4—Reflect.

The optical transmission coefficient of the directional coupler is determined as [32,37]
(5)$$T_{through}=\frac{P_{through}}{P_{input}}=\cos^{2}\left(\frac{\pi \Delta n_{eff}}{\lambda }L\right),$$
where *P_through_* is the output optical power at the through port, *P_input_* is the input optical power at the input port, *λ* is the wavelength, *L* is the coupling length, and ∆*n_eff_* is the difference between the effective indices of the even and odd modes.

At the first stage, the method of the Finite Difference Eigenmode (FDE) was used to obtain the dependencies of the optical transmission coefficient of the OMT on the coupling length *L* and air gap (Figure 11). The calculations using the FDE method implied the cross-section of two coupled waveguides with dimensions of 350 nm × 850 nm and a wavelength of 1550 nm, with air gap between them [35]. In mathematical terms, the FDE method calculates the effective indices of the even and odd modes of the two coupled waveguides. The obtained data can be used to calculate the difference between the effective indices of the even and odd modes ∆*n_eff_* in Equation (5) or to obtain data on the transmission coefficient in a selected waveguide (waveguide of the through port in our case) for a necessary coupling length immediately in the solver.

Evidently, from the plot (Figure 11), the increased gap increases the critical coupling length, which corresponds to the total migration of the optical power from one waveguide into another. In addition, at a fixed coupling length, there are several local minima and maxima of the optical power in the through port, which is explained by the fact that at small gaps, optical power manages to migrate to the drop port and back. Figure 12 shows power maps of the input coupling structure with the gap ranging from 105 to 500 nm.

The linear region enlarges and the slope diminishes with an increasing gap and extended power exchange period, which is due to smaller overlap between the waveguiding modes. In work [35], we presented the dependencies of the difference between the effective indices of the even and off modes of the gap that is inserted into Equation (5). More details on the selection of the displacement type of the moving waveguide and the optical measuring transducer working principle can be found elsewhere [9,32].

To increase the working gap in further studies, we have selected a directional coupler with a coupling length of 100 μm. In Figure 12, this point corresponds to the top point of the left waveguide (through port).

At the second stage, the Finite Difference Time Domain (FDTD) method was used to obtain the dependencies of all the S-parameters of the OMT on the gap value. Figure 13 presents the plot for *S*21 (through port) for gap range from 100 nm to 600 nm.

Point 360 nm (Figure 13) is the working gap that corresponds to an optical transmission coefficient (OTC) of *T_through_* ≈ 0.5 and should be set by the OTPS.

The received model of the OMT contains the dependence of the S-parameters on the gap values from 100 to 1000 nm for a wavelength range from 1500 to 1600 nm. Figure 14 presents a single point of the received S-parameters at a selected working gap of 360 nm. Evidently, from Figure 14a, S21 and S31 intersect at a wavelength of 1550 nm, while the transmission coefficient is less than 0.5, which is conditioned by the losses in the waveguides and cross-coupling losses.

The working section at which the OMT can measure displacements (Figure 13) that are proportional to the acting acceleration along the sensitivity axis lies in the range from (360_−80_) nm to (360^+140^) nm, where the OTC varies from 0.002 to 0.906. The OTC changes asymmetrically, which should be taken into account during the output signal processing. To linearize the working section of the OTC, the dynamic range should be reduced to a range from (339_−39_) nm to (339^+39^) nm, where the OTC varies in a range of 0.32 ± 0.27.

The transmission coefficient of the OMT was determined as
(6)$$K_{o}=\frac{T_{opt}^{\max}-T_{opt}^{\min}}{G_{\max}-G_{\min}},$$
where $T_{opt}^{\max},\;T_{opt}^{\min}$ are the maximum and minimum optical power at the coupler output; and $G_{\max},\;G_{\min}$ are the maximum and minimum gaps between the waveguides. For the obtained linearized characteristics of the OMT, the calculated transmission coefficient amounted to 6.779 × 10^6^ m^−1^.

## 4. Fabrication of the Accelerometer

The experimental MOEM accelerometer was fabricated on a four-inch SOI wafer according to the technological process presented in Figure 15 (shown in a simplified form).

The process of the accelerometer fabrication included the following steps. First, the terminal pads were metalized using Ti/Au films via the method of electron-beam sputtering through a two-layer photoresistive LOR5B/AZ1505 mask. Then, optical waveguides on the basis of SiO_2_/Si_3_N_4_/SiO_2_ films were formed using the methods of plasma chemical deposition and plasma chemical etching (Figure 15a). The results are presented in Figure 16.

At the next stage, the fixed and moving parts of the accelerometer’s MSE were formed (Figure 15b). This stage of the fabrication flow included electric insulation of the moving and fixed parts of the accelerometer, and formation of the comb electrode structure and functional gap between the waveguides. Before deep etching of the SOI wafer, anisotropic etching of the 4 μm SiO_2_ film through a metallic 100-nm Al film was performed.

Then, the wafer with the MSE structure was bonded to the temporary carrier wafer (Figure 15c). The bonding was performed using a photoresist HT10.11 that was applied to the carrier wafer via centrifugation with consequent application to the device wafer and stage-wise drying at +190 °C. Then, the reverse side of the device wafer was thinned to a thickness of 200 μm via chemical–mechanical polishing (Figure 15d). Then, liquid etching of silicon to a depth of 170 μm (Figure 15e) was performed through a dielectric SiO_2_ mask with a thickness of 100 nm. At the next stage, the device wafer with the formed functional elements of the accelerometer’s MSE was bonded to the base wafer (Figure 15f). The bonding was performed using a Benzocyclobutene (BCB) polymer adhesive. At the last stage, the photoresist HT10.11 was lifted off in toluene and the carrier wafer was removed (Figure 15g). The presented technology allowed for fabricating the prototype of the accelerometer’s MSE (Figure 17).

## 5. Experimental

### 5.1. Setup

The accelerometer’s prototype was tested by determining the output optical power of the OMT under acceleration. Figure 18 depicts the experimental setup used to record the optical power at the output of the accelerometer’s prototype. The experimental setup was mounted on a vibration isolation table.

Accelerometer prototype 1 was mounted via vacuum on manual triaxial positioner 2, which was aligned with the horizontal plane. Fiber holder 5 for edge coupling was mounted on positioners 3 (six-axis automatic Luminos U6 Ultra XYZ/RYP positioner, Ottawa, ON, Canada) and 4 (six-axial manual Luminos I6000 XYZ/RYP positioner, Ottawa, ON, Canada). A single-mode lensed fiber SMF28 with a mode diameter of 3 μm was used. Into the end of the accelerometer’s fixed waveguide, optical radiation with a wavelength range of 1527–1563 nm generated via laser 6 (Keysight N7714a, Santa Rosa, CA, USA) was fed. The optical fiber was positioned to the fixed waveguide by automatic positioner 3 either manually using joystick 7 or using software installed on personal computer 8. The output optical fiber was aligned respective to the outlet edge of the accelerometer’s prototype using manual positioner 4. To obtain the quantitative characteristics of the optical power output through the fixed waveguide of the accelerometer’s prototype, optical radiation power meter 9 (Keysight N 7744c, Santa Rosa, CA, USA) was used.

### 5.2. Testing of the Accelerometer

All the prototypes of the accelerometer possessed a system of comb electrodes (positioning system) that solved two tasks: working gap setup and simulation of acting acceleration. For this purpose, sources of electric voltage 10 and 11 (KEYSIGHT E36313A and KEYSIGHT E3631A, Santa Rosa, CA, USA) were used. The accelerometer prototypes were soldered into the housing for ease of voltage supply to the electrode structures (Figure 18).

At the first stage, the working gap was set and the gravity-induced MSE displacement (along the Z axis) was compensated for. For this purpose, voltage was applied to the comb structures of the MSE’s positioning system as per the scheme presented in Figure 19.

To compensate for gravity, voltage *U*_1_—equal to 0.8 of the calculated value (13.5 V)—was fed from an E36313A voltage source to the electrodes (Figure 19). This generated the electrostatic force that displaced the MSE along the Z axis. To find the actual voltage *U*_1_, we increased its value step by step and at each step altered the voltage *U*_2_ to reach the minimal power at the optical power meter, which would mean that the waveguides lie in a single plane and the absolute maximum of optical power is transferred to the drop port and disperses. For further tests, voltage *U*_1_ was fixed.

Under voltage *U*_2_ at the electrode structures, the MSE was displacing along the Y axis, which was altering the gap between the waveguides and, hence, the output optical power. From the E36313A power source, the sum of voltages *U*_1_ and *U*_2_ was fed to electrode structures *C*1*_ls_* and *C*1*_rs_*. With changing voltage *U*_2_, the Pout was recorded, and the dependence of output power on voltage *U*_2_ was plotted. On the obtained plot, the most linear section was selected, and voltage *U*_2_ was set so that it corresponded to the middle of this section. This voltage corresponded to the working gap at which further acceleration testing was performed.

Three prototypes of accelerometers were selected for the tests, for which the voltages necessary for the working gap setup and gravity compensation amounted to the following: *U*_1*acc*1_ = 14.769 V; *U*_1*acc*2_ = 14.391 V; *U*_1*acc*3_ = 13.696 V; *U*_2*acc*1_ = 8.831 V; *U*_2*acc*2_ = 8.462 V; *U*_2*acc*3_ = 7.891 V.

The tests involved the simulation of the acceleration with the applied voltage measured using the accelerometer. This required comb electrodes *C*2*_l_* and *C*2*_r_* (Figure 9) to create electrostatic force *F_e_* equal to the inertia force at a corresponding acceleration. This electrostatic force induced the same displacement of the MSE as the measured acceleration. Then, the simulated acceleration could be determined as
(7)$$a_{y}=\frac{F_{e}}{M}=\frac{\partial C_{i}}{\partial y}\frac{1}{M}U_{3}^{2},$$
where *C_i_* is the capacity of the electrode structures, F; and *U*_3_ is the voltage at electrodes *C*2*_l_* and *C*2*_r_*, V. The values of the simulated acceleration and corresponding calculated voltages are presented in Figure 20.

Figure 21, Figure 22 and Figure 23 present the experimental dependencies of the output optical power of the OMT on the acceleration in different ranges for the three prototypes of accelerometers. Solid lines denote linear approximations of the accelerometer’s transmission coefficient in the average value at its most linear section limited in a range from minus 0.01 g to plus 0.01 g and correspond to Function (8)
(8)$$P_{outapprox}=\left|P_{out0}+k\times g\right|,$$
where *P_out_*_0_ is the output power at g = 0, μW; and *k* is the linear approximation coefficient, or power gain.

Following the experimental data, the less was the measured range, the less was the nonlinearity of the obtained characteristic. The nonlinearity was calculated as
(9)$$\mathrm{Nonlinearity}=\frac{P_{out}+P_{outapprox}}{P_{outapprox}}\cdot 100\%,$$
where *P_out_* is the output power, μW.

For the dynamic range from minus 0.01 g to plus 0.01 g, the average nonlinearity of the accelerometers’ characteristics ranged from 0.7% to 1.62%. For the maximum dynamic range from minus 0.015 g to plus 0.05 g, the nonlinearity ranged from 2.34% to 2.9%, having the maximum deviation at the edges of the regions. The power gain *k* of the three prototypes of accelerometers varied from 12.321 mW/g to 26.472 mW/g. When using a standard photodiode with a transmission coefficient of 0.7 A/W instead of an optical power meter, the transmission coefficient of the three prototypes of accelerometers with photodiode outlet ranged from 8.6247 mA/g to 18.53 mA/g. The difference in sensitivity is due to the different eigenfrequencies of the prototypes. The actual sensitivity threshold amounted to 2 μg, which is evident from the optical power meter data (Figure 23). According to the experimental results, the fluctuations in the power source voltages and residual vibration of the base prevented us from unambiguously identifying voltage *U*_2_ to simulate the acceleration induced by those factors. The averaged data from the optical power meter over a period from 10 ms to 1 s also prohibited reaching a definite conclusion on the sensitivity threshold below 2 μg.

## 6. Conclusions

We have designed, fabricated and tested an MOEM accelerometer consisting of an MSE and an OMT based on the tunneling effect. An original mechanical structure of the MSE for the MOEM accelerometer was developed; the necessary parameters of the MSE were analyzed and selected. Thanks to the acceleration-sensitive MSE and the optical measuring transducer based on a directional coupler, the accelerometer can achieve the high sensitivity and resolution of the acceleration measurement. The experimental results showed that the MOEM accelerometer of the proposed design possesses a sensitivity threshold of 2 μg and a sensitivity to acceleration from 8.6247 mA/g to 18.53 mA/g; the intrinsic noise of the accelerometer did not exceed 2 μg/√Hz at 1 Hz. The linear section of the output power dependence on the transmission coefficient for all the prototypes did not exceed 0.01 g, which is a fairly small value for practical application. One of the ways to reduce the sensitivity threshold and increase the dynamic range is to implement a displacement feedback system, which is to be studied in further works.

The most promising application spheres of the proposed MOEM accelerometer under development are seismic exploration and inclination measurement in systems for critical structure monitoring.

## Figures and Tables

**Figure 1 sensors-24-00765-f001:**
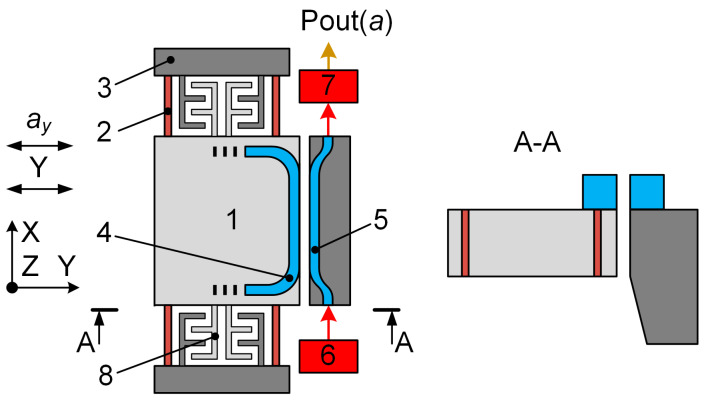
Functional scheme of an MOEM accelerometer with an OMT.

**Figure 2 sensors-24-00765-f002:**
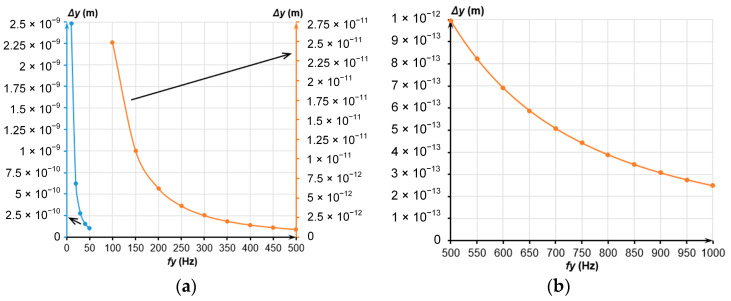
MSE displacements along the Y axis; (**a**) Frequency range from 10 to 500 Hz; (**b**) Frequency range from 500 to 1000 Hz.

**Figure 3 sensors-24-00765-f003:**
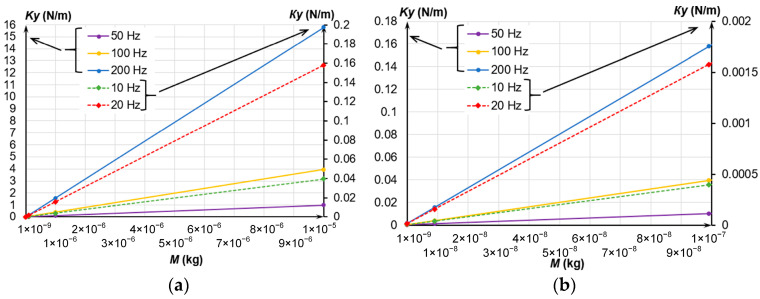
Dependence of the spring suspension stiffness on the MSE’s mass at different frequencies; (**a**) Mass change from 1 × 10^−9^ to 1 × 10^−5^ kg; (**b**) Mass change from 1 × 10^−9^ to 1 × 10^−7^ kg.

**Figure 4 sensors-24-00765-f004:**
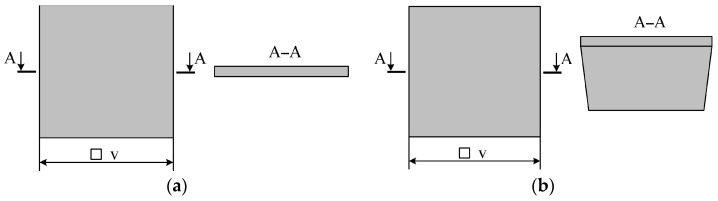
Types of MSEs (spring suspension is not given); (**a**) Type 1 MSE; (**b**) Type 2 MSE.

**Figure 5 sensors-24-00765-f005:**
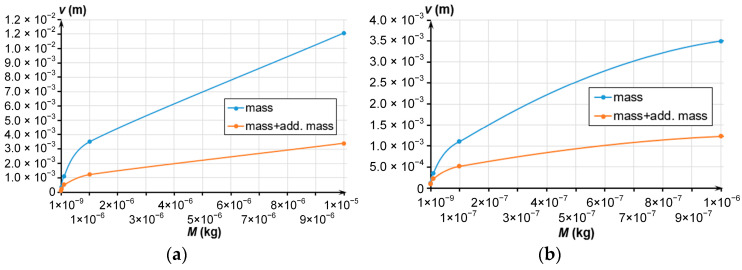
Dependence of an MSE’s dimensions on its mass; (**a**) Mass change from 1 × 10^−8^ to 1 × 10^−5^ kg; (**b**) Mass change from 1 × 10^−8^ to 1 × 10^−6^ kg.

**Figure 6 sensors-24-00765-f006:**
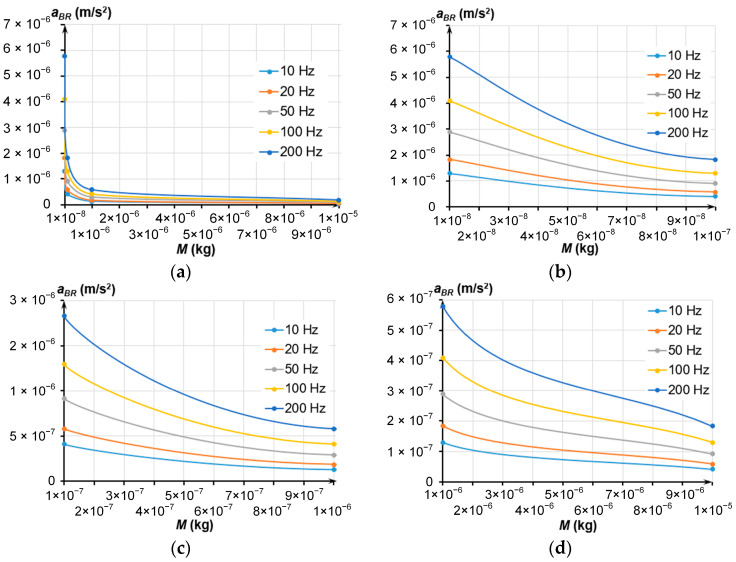
Dependencies of the *a_BR_* on the MSE’s mass; (**a**) Mass change from 1 × 10^−8^ to 1 × 10^−5^ kg; (**b**) Mass change from 1 × 10^−8^ to 1 × 10^−7^ kg; (**c**) Mass change from 1 × 10^−7^ to 1 × 10^−6^ kg; (**d**) Mass change from 1 × 10^−6^ to 1 × 10^−5^ kg.

**Figure 7 sensors-24-00765-f007:**
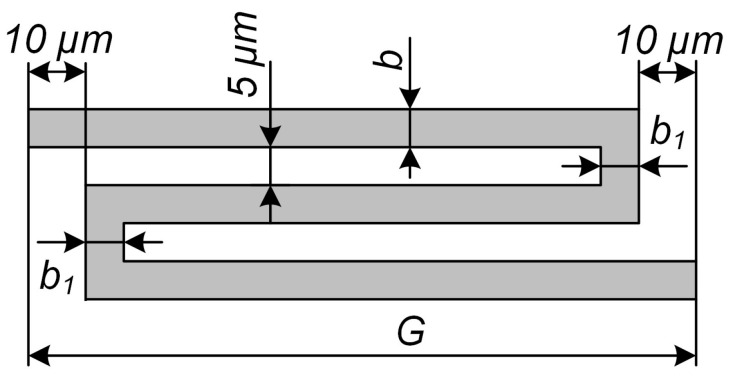
Spring-type elastic elements of the suspension.

**Figure 8 sensors-24-00765-f008:**
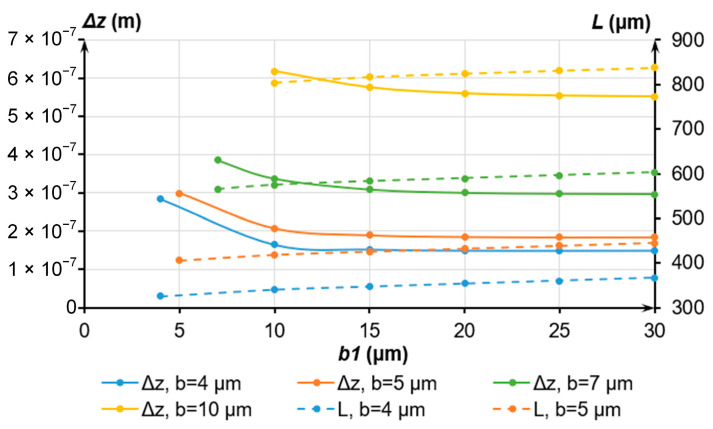
Dependence of the displacement Δ*z* and length *G* of the elastic elements on the dimensions *b* and *b*_1_.

**Figure 9 sensors-24-00765-f009:**
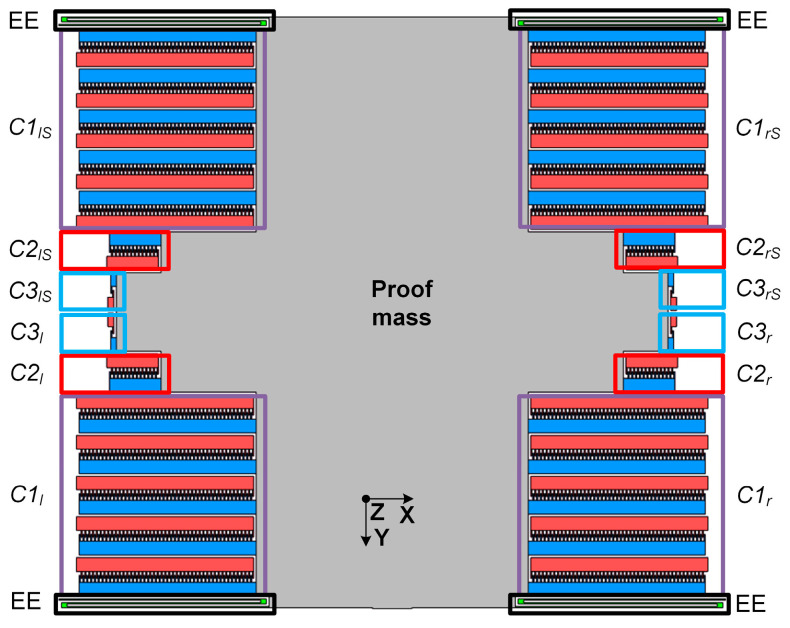
Design of the accelerometer’s MSE (optical waveguides are not shown).

**Figure 10 sensors-24-00765-f010:**
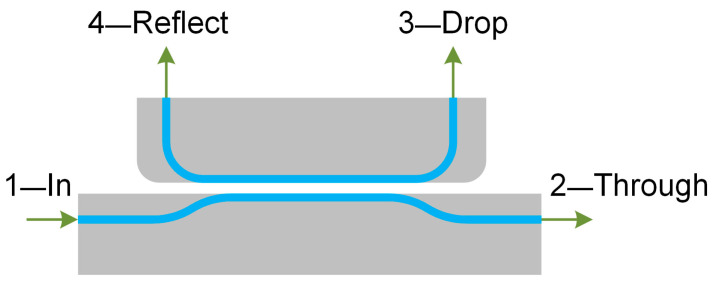
Model of the OMT’s directional coupler.

**Figure 11 sensors-24-00765-f011:**
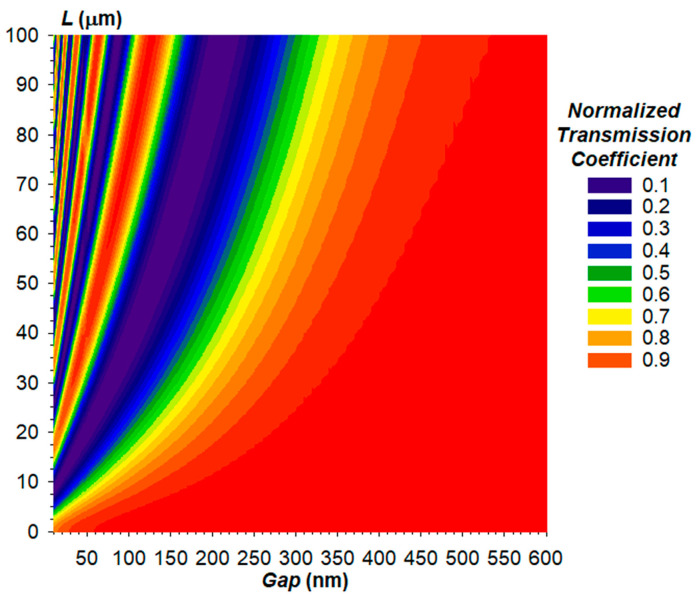
Dependence of the optical transmission coefficient of the directional coupler’s through port on the coupling length and gap at a wavelength of 1550 nm.

**Figure 12 sensors-24-00765-f012:**
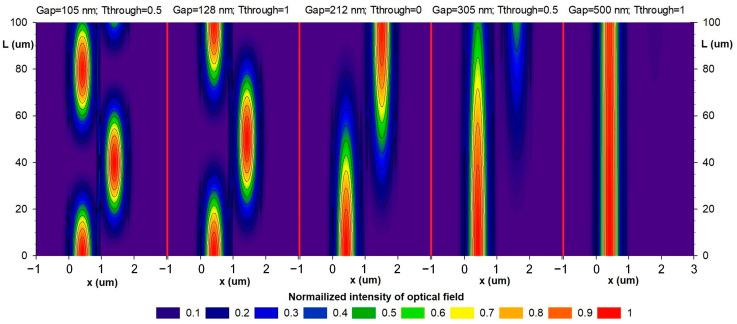
Power maps at different gaps obtained via the FDE method.

**Figure 13 sensors-24-00765-f013:**
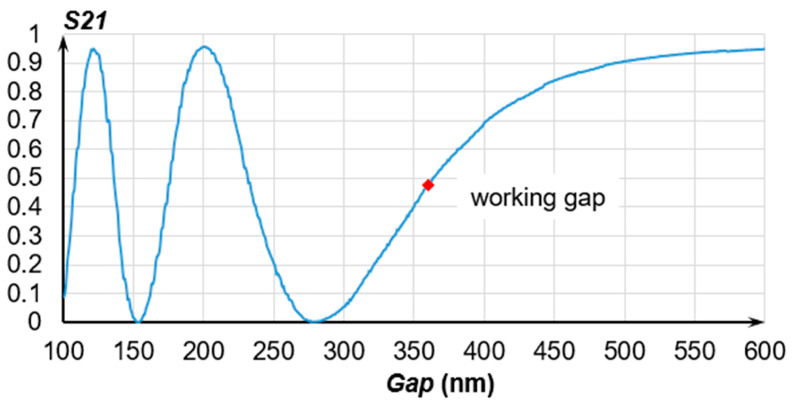
Dependence of the OMT’s S21 on the gap at a wavelength of 1550 nm obtained via the FDTD method.

**Figure 14 sensors-24-00765-f014:**
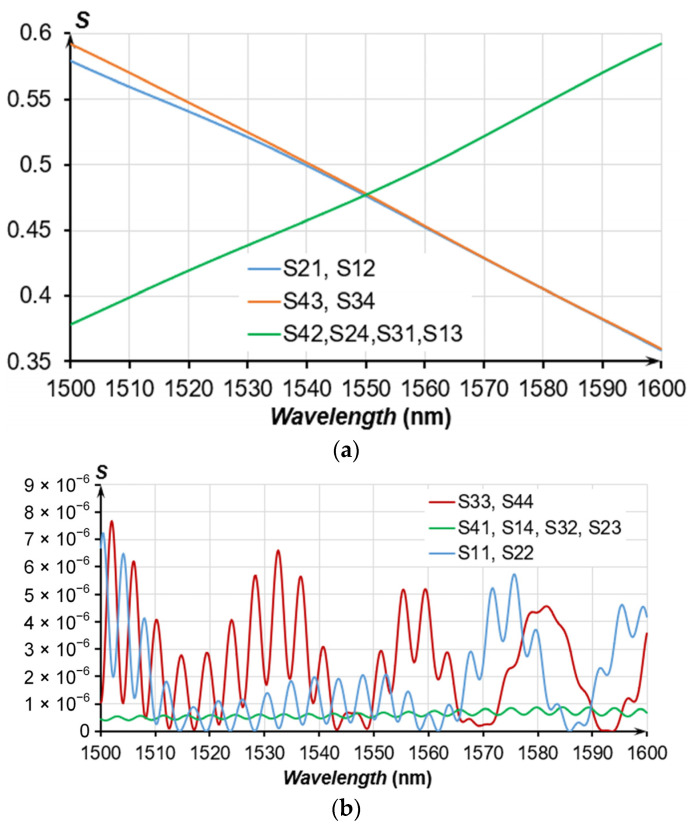
S-parameters of the OMT for a gap of 360 nm; the waveguide cross-section is 350 nm × 850 nm and the coupling length is 100 µm; (**a**) S21, S31, S42, S43, S12, S13, S24, S34; (**b**) S11, S22, S33, S44, S41, S14, S32, S23.

**Figure 15 sensors-24-00765-f015:**
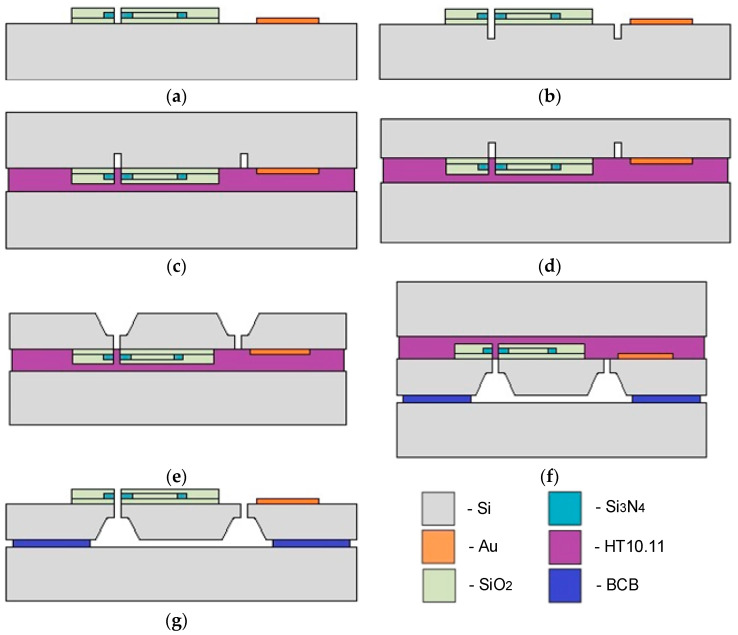
MSE fabrication process flow: (**a**)—formation of the optical waveguides and metallic plates; (**b**)—separation of functional elements; (**c**)—bonding to the carrier wafer; (**d**)—thinning of the device wafer; (**e**)—etching of the reverse side of the device wafer; (**f**)—bonding to the base wafer; and (**g**)—debonding of the carrier wafer.

**Figure 16 sensors-24-00765-f016:**
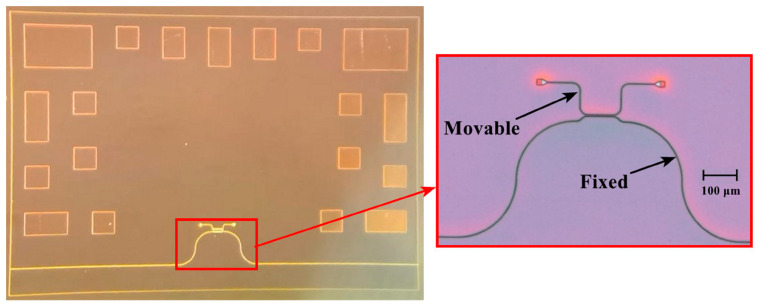
Formed optical waveguides.

**Figure 17 sensors-24-00765-f017:**
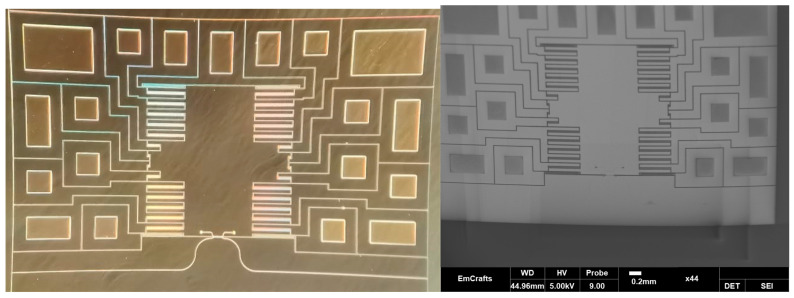
Prototype of the accelerometer’s MSE.

**Figure 18 sensors-24-00765-f018:**
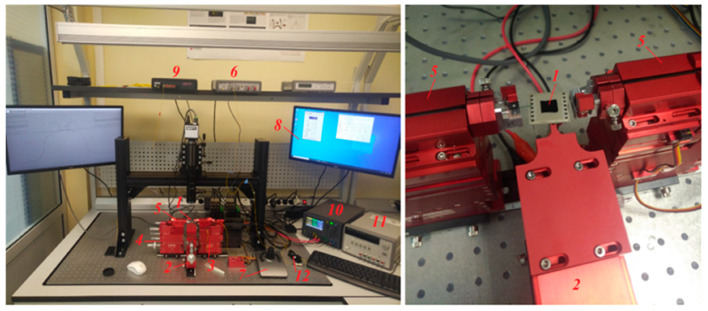
Experimental setup: 1—accelerometer prototype; 2—triaxial positioner; 3—six-axis automatic positioner; 4—six-axis manual positioner; 5—fiber holder; 6—laser; 7—joystick; 8—personal computer; 9—power meter; 10—sources of electric voltage; 11—sources of electric voltage; 12—polarizer.

**Figure 19 sensors-24-00765-f019:**
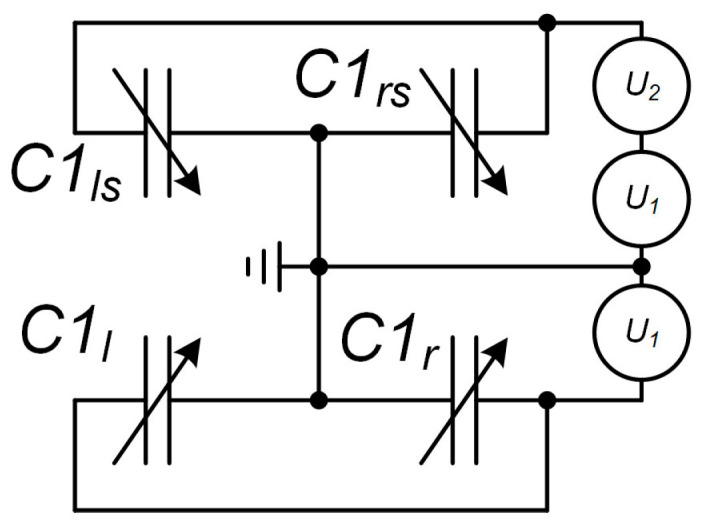
Connection scheme of the positioning system electrodes.

**Figure 20 sensors-24-00765-f020:**
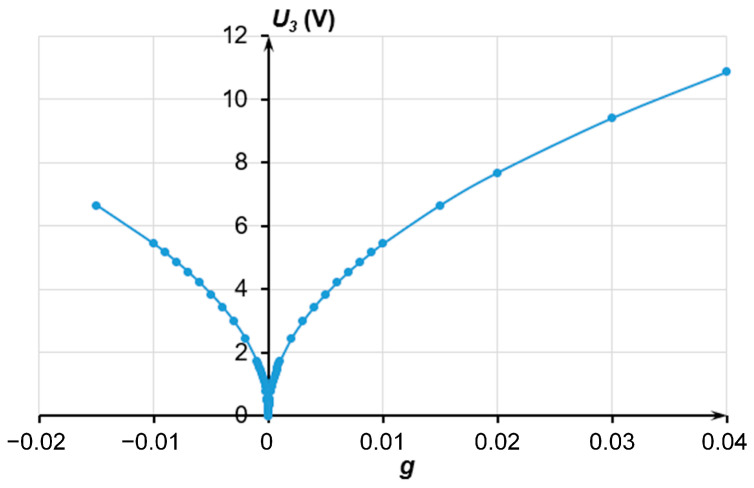
Dependence of the simulated acceleration on the calculated voltage *U*_3_.

**Figure 21 sensors-24-00765-f021:**
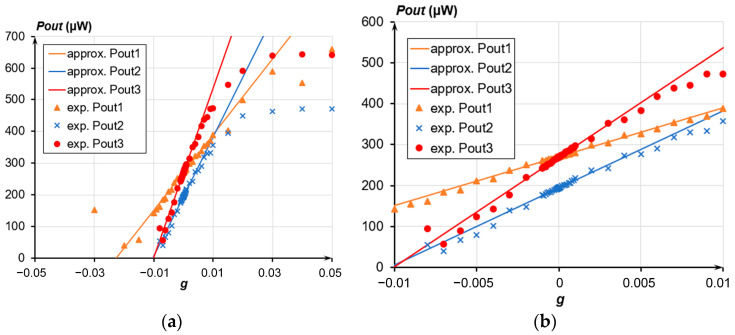
Dependence of the accelerometer’s output optical power on the acceleration; (**a**) Range from minus 0.03 g to plus 0.05 g; (**b**) Range from minus 0.01 g to plus 0.01 g.

**Figure 22 sensors-24-00765-f022:**
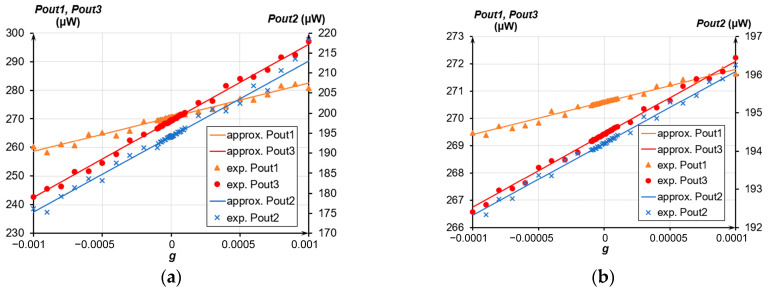
Dependence of the accelerometer’s output optical power on the acceleration; (**a**) Range from minus 0.001 g to plus 0.001 g; (**b**) Range from minus 0.0001 g to plus 0.0001 g.

**Figure 23 sensors-24-00765-f023:**
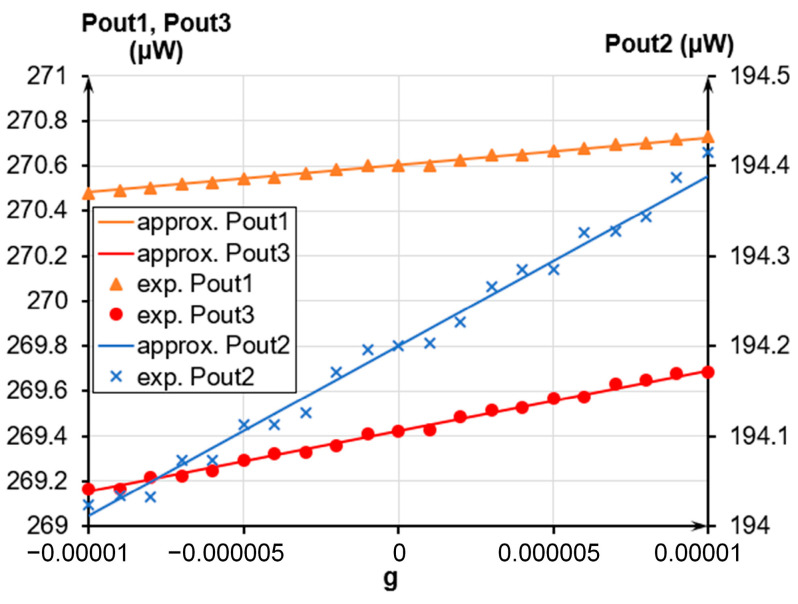
Dependence of the accelerometer’s output optical power on the acceleration in a range from minus 0.00001 g to plus 0.00001 g.

**Table 1 sensors-24-00765-t001:** Summary of the characteristics of MOEMS accelerometers.

Device Type	Sensitivity	Eigenfrequency	Intrinsic Noise	Measuring Range	Bandwidth
Photonic-crystal nanocavity [14]	10 µg Hz^−1/2^	20 kHz	-	-	50 dB
Sub-wavelength gratings [15]	2033 nm/g	379 Hz	-	0.12 g	-
Interferometric position sensor [16]	-	From 80 Hz to 1 kHz	40 ng/rt Hz	-	85 dB at 40 Hz
Fiber Bragg grating [17]	~100 pm/g cross-axis anti-interference degree < 5%	(10–120) Hz	-	-	-
Fiber grating [18]	-	-	-	-	Up to 200 Hz
Electron tunneling transducers [21]	-	-	20 ng/Hz	-	5 Hz–1.5 kHz
Ring resonators [22]	18.9/g	-	4.874 μg	From −23.5 g to 29.4 g	-
Micro-grating-based [24]	169 μm/g 60 V/g	-	15 ng/√Hz for 1 Hz	-	-
Fabry–Pérot interferometer (FPI) [25]	(1.022–1.029) mV/(m/s^2^)	1274 Hz	-	7 g	-
Fiber-free optical [26]	0.156 mA/g, resolution of 56.2 µG	-	-	-	
Fabry–Perot (FP) cavities [27]	X-axis—309 μg Y-axis—313 μg	X-axis—1382.5 Hz Y-axis—1398.6 Hz	-	1 g	-
Michelson interferometer structure [28]	3.638 nm/g	1742.2 Hz	-	±500 g	-
Talbot effect [29]	0.14 μm/g 0.74 V/g	1878.9 Hz	2.0 mg	-	-
Intensity modulation of light [30]	600 nm/g	560 Hz	-	3 g	-
Photonic crystal [31]	0.0750 nm/g	17.7 kHz	-	±200 g	-
Optical tunneling effect [32]	(6.25 × 10^6^ m^−1^)	(10–200) Hz	-	-	-
Optical tunneling effect [33]	9 pm/g	-	-	±130 g	0–1500 Hz

**Table 2 sensors-24-00765-t002:** Length of the spring suspension at different eigenfrequencies.

	*M*, kg	1.00 × 10^−8^	1.00 × 10^−7^	1.00 × 10^−6^	1.00 × 10^−5^
*f_y_*, Hz	*f_z_*, Hz	*G*, μm			
10	87.5	23,510	10,910	5060	2370
20	175	14,770	6860	3200	1505
50	437.5	8040	3760	1760	836
100	875	5094	2374	1111	539
200	1750.04	3210	1509	720.3	353.7

**Table 3 sensors-24-00765-t003:** Displacement of the MSE along the Z axis at different mass and frequency.

	*M*, kg	1.00 × 10^−8^	1.00 × 10^−7^	1.00 × 10^−6^	1.00 × 10^−5^
*f_y_*, Hz	*f_z_*, Hz	Δ*z*, μm			
10	87.5	32.8831	32.7483	32.5152	33.2363
20	175	8.16708	8.12145	8.20043	8.52342
50	437.5	1.3078	1.32746	1.35832	1.5075
100	875	0.3331137	0.333466	0.355366	0.440739
200	1750.04	0.0826125	0.085916	0.0983281	0.149845

## Data Availability

The data presented in this study are openly available in FigShare at https://figshare.com/articles/dataset/Calculated_and_experimental_data_for_figures/25053299/1 (accessed on 19 December 2023).

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
