# Peer review of "The Design, Modeling and Experimental Investigation of a Micro-G Microoptoelectromechanical Accelerometer with an Optical Tunneling Measuring Transducer"

_sensors, 2024, doi:10.3390/s24030765_

Round 1
Reviewer 1 Report
Comments and Suggestions for Authors
The manuscript "The design, modeling and experimental investigation of a micro-g MOEM-accelerometer with an optical tunneling measuring transducer" submitted to Sensors, shows an interesting idea. In this work, the theory, modeling and experimental demonstration of the MOEM-accelerometer based on the optical tunneling effect are systematically offered, showing high sensitivity and resolution of acceleration measurement due to its acceleration-sensitive chip. However, some questions still need to be addressed before publication.
1. According to the work, the abstract can not reflect the importance of the work, such as the challenges of the fields and how the authors deal with it. This part should be highlighted.
2. In terms of the fabrication of an accelerometer, is it possible to use solution-processed methods such as transfer printing and inkjet printing? And how about a flexible accelerometer? (Adv. Mater. 2023, 2210385, https://doi.org/10.1016/j.matt.2023.05.043)
3. Some format and language expressions in the manuscript need to be carefully checked. For example, the case format of the first letter of a word in Table 1 and 2 and Figure 13.
4. The numbers of the Figures in the work are too many. Some Figures can be integrated for better representation.

Please see comments.
Author Response
Thank you for an objective peer-review of our article and the comments provided; this allowed us to improve the presentation of our research results.
- According to the work, the abstract can not reflect the importance of the work, such as the challenges of the fields and how the authors deal with it. This part should be highlighted.
Answer: The abstract has been corrected.
- In terms of the fabrication of an accelerometer, is it possible to use solution-processed methods such as transfer printing and inkjet printing? And how about a flexible accelerometer? (Adv. Mater. 2023, 2210385, https://doi.org/10.1016/j.matt.2023.05.043)
Answer: Any materials are suitable for fabricating optical waveguides by any applicable methods. In principle, an optical accelerometer with optical waveguides on the basis of Perovskite micro-/nanoarchitecture can be also fabricated. Moreover, the most promising method is the integration of laser, passive elements and photodiode in a single fabrication cycle without assembly operations. However, the issues of integration of transfer printing and inkjet printing technologies with more conventional processes of deep silicon etching (Bosch-process) that are used for proof mass fabrication require further investigation.
We believe, flexible accelerometers can be fabricated and tested. However, their practical application will be restricted, because the optical waveguide, in this case, will represent the proof mass and will be subjected to bending, which will immediately affect the reliability, induce internal losses and deteriorate the repeatability of the results. In our case, the optical and mechanical parts are functionally separated and do not suffer the effects of crosstalk.
We added a sentence in the introduction (line 49) to emphasize the importance of research in the field you mentioned: Particularly promising are MOEMS-technologies that can integrate lasers, photodiodes, passive elements and mechanics in a single fabrication cycle [23].
- Some format and language expressions in the manuscript need to be carefully checked. For example, the case format of the first letter of a word in Table 1 and 2 and Figure 13.
Answer. We have corrected the letter case in all tables and figures.
- The numbers of the Figures in the work are too many. Some Figures can be integrated for better representation.
Answer. We have removed some figures and combined some other ones for better presentation of the results
Reviewer 2 Report
Comments and Suggestions for Authors
The authors proposed a micro-optoelectromechanical accelerometer with an optical measuring transducer built on the optical tunneling principle. The characteristics of the microelectromechanical sensing element is analyzed according to the design indicator of 1 µg m/s2. The experiments show a reasonable sensitivity threshold of the 2 µg. The authours use a method based on optical coupling between micro-/nanowaveguides, which has already been reported before (C. Xin et al. Scientific Reports 2022, 12, 21697; A. K. Shotorban et al. IET Circuits, Devices & Systems 2019, 13, 1102). The quality of presentation is low. And the results in simulation and experiment are not explained and discussed sufficiently. In summary, I don’t think this manuscript can meet the high quality of Sensors. The authors may consider the following questions:
1. The english language is greatly suggested to be improved carefully. There are plenty mistakes (e.g., tense). The name of the axis are located at different positions in different figures.
2. There are too many abbreviations, which makes the manuscript less readable. Some short terms like 'elastic elements' and 'proof mass' are not necessary to be used as abbreviations. Or, an abbreviations' list is suggested.
3. The Q value used in the manuscript should be stated and analyzed.
4. In section 2.1 the last paragraph (‘the gap between the waveguides was adjusted by an optical transducer positioning system (OTPS) that was presented by comb electrode structure (CES) 8.’) (line 162) has shown that the technology of OTPS can be used to adjusted micro-nano displacement precisely. Several references about applications of OTPS technology are suggested to cite following this sentence.
5. In Fig. 4 and Eq. 3 (“dimension ‘a’”) is suggested to change into another symbol, avoiding confusion with acceleration ‘a’.
6. Line 233 check the relevant reference.
7. Line 327 Finite Difference Eigenmode (FDE)
8. Line 329 350×850 nm^2
9. The simulated results shown in Fig.12-13 should be expained and discussed. For example, in Fig.12, the curve shows a oscillating property with gap less than ~300nm. However, as gap increasing over ~300nm, the oscillation disappears. And in Fig.13, the reason and influence on the design of the sensor of the simulated curve should also be discussed.
10. In section 5.2 the penultimate paragraph (line 453), what is the definition of ('its most linear section.')? what method is used to quantify it? least-square method? or other methods? Similarly, what is the definition of ('nonlinerity') in section 5.2 the last paragraph (line 469)?
11. In Fig. 4, what's the advantage and disadvantage of the two types of MSEs, which may be the reason why these two types are put together to be compared.
12. In page 7 the last paragraph (line 241), why (' the increase of its linear dimensions decreases its proof mass stiffness')? whether it is based on Eq. 3, the increase in 'a' decreases the 'h', thus leading ot the stiffness decreasing, when 'm' is a constant?
Comments on the Quality of English LanguageThe quality of presentation is low. And the english language is greatly suggested to be improved carefully. There are plenty mistakes (e.g., tense).
Author Response
Thank you for an objective peer-review of our article and the comments provided; this allowed us to improve the presentation of our research results.
Regarding the works of Xin, we are familiar with their work. We already had a reference to their work of 2019 and have added a new reference to their article of 2022 during the improvement of our work. Shatorban has been already mentioned in our work. We should note that their works consider only modeling without further fabrication, while our work pays much attention to the mechanical part of the transducer and it fabrication. We have also described in detail the experimental part of the study regarding fabricated MOEM prototypes. Xin has an outstanding article in optics; however, we have performed similar investigation earlier, which is indicated by the references to our works [32, 34]. In the answer to question 9 we presented an extensive explanation and also have improved corresponding part in our work.
All answers to your suggestions and comments are given below.
- The english language is greatly suggested to be improved carefully. There are plenty mistakes (e.g., tense). The name of the axis are located at different positions in different figures.
Answer. We have revised the style and grammar of the text. We hope it got better. The names of axes in all figures have been corrected.
- There are too many abbreviations, which makes the manuscript less readable. Some short terms like 'elastic elements' and 'proof mass' are not necessary to be used as abbreviations. Or, an abbreviations' list is suggested.
Answer. We have removed abbreviations of such terms as spring suspension, elastic elements, proof mass, comb electrode structure.
- The Q value used in the manuscript should be stated and analyzed.
Answer. To reduce the transient time, when measuring acceleration, the Q-factor should be minimal. For our case, Q-factor was equal to 20. This information was added in line 199.
- In section 2.1 the last paragraph (‘the gap between the waveguides was adjusted by an optical transducer positioning system (OTPS) that was presented by comb electrode structure (CES) 8.’) (line 162) has shown that the technology of OTPS can be used to adjusted micro-nano displacement precisely. Several references about applications of OTPS technology are suggested to cite following this sentence.
Answer. The implementation of the OTPS method was invented by the very authors of the presented article, because there are limitations in terms of the aspect ratio in Bosch Process. Aspect ratio in our case can be calculated as follows: proof mass height/initial working initial gap=35 um/360 nm ~100. OTPS is a capacitive actuator that is widely used in feedback systems of various MEMS accelerometers and gyroscopes. This system is described in more details in section 2.2.3 and in our article [35] that is among the references.
To improve the presentation of material, we have added a direct reference to the article and section 2.2.3 in line 170. We have also mentioned inaccurate title of the section 2.2.3 and have corrected it to OTPS. In the beginning of section 2.2.3 (line 303) we have added a sentence with an additional reference to the source describing actuator systems [36].
- In Fig. 4 and Eq. 3 (“dimension ‘a’”) is suggested to change into another symbol, avoiding confusion with acceleration ‘a’.
Answer. In Figures 4 and 5 and in equation 3 we have substituted a with v.
- Line 233 check the relevant reference.
Agree. Corrected the typo in line 241.
- Line 327 Finite Difference Eigenmode (FDE)
Agree. Corrected the typo in line 340.
- Line 329 350×850 nm^2
Answer. We meant not the area of the waveguide, but its dimensions, i.e., 350 nm in height and 850 nm in width. We have corrected line 343.
- The simulated results shown in Fig.12-13 should be expained and discussed. For example, in Fig.12, the curve shows a oscillating property with gap less than ~300nm. However, as gap increasing over ~300nm, the oscillation disappears. And in Fig.13, the reason and influence on the design of the sensor of the simulated curve should also be discussed.
Answer. Figure 12 (13 in current numeration) is a particular case of OMT transmission coefficient dependence on the gap at a fixed coupling length of 100 µm (Figure 11) that was obtained by the FDTD method at a wavelength of 1500 nm. The data received by FDTD are more factual, because they consider a complete 3D-model rather than only the cross-section of coupled waveguides as in the case of the FDE method (Figure 11).
In mathematical terms, the FDE method calculates the effective indices of even and odd modes of coupled waveguides that can be used either in Equation (5), or immediately in the solver to obtain data on the transmission coefficient in a selected waveguide (through port waveguide, in our case). In works [32, 35] we have presented an extensive description of the measuring transducer and the dependence of the difference between the effective indices of even and odd modes of the gap that is used in Equation (5). In the presented article, these dependencies would be irrelevant, because the dependence of OMT on the gap was eventually obtained by the FDTD method (Figure 13).
The oscillating behavior of the curve in Figure 12 (13 in current numeration) with the gap below ~300 nm is explained by the following. At decreasing gap at a fixed coupling length, the optical power manages to migrate to the drop port and back. The smaller the gap, the lesser the critical coupling length that corresponds to complete migration of the optical power from one waveguide to another. Since the length of the directional coupler is fixed (100 µm), then the smaller the gap, the more migrations of the optical radiation occur.
For the present work, Figure 13 (14 in current numeration) was given to show that for further modeling of the measuring transducer, a complete matrix of all S-parameters was used with due regard to the “parasitic” effects (Fig. 14b), rather than an approximate value that was calculated only for the through port by the FDE method. Therefore, a reader understands that the transmission coefficient also depends on the wavelength, which is evident from Equation (5). Besides, there are works studying the dependence of the source wavelength instability on the optical transmission coefficient, which specifically requires a complete model of S-parameters. In current work, we did not present these data, because in the experiments we have used a highly stable laser source; moreover, we tried to keep the article more concise. The mentioned studies will be presented in further publications.
We appreciate all your comments and, hence, supplemented descriptions in lines 340-349, 353-358, 361-372, 376-384 and 390-393. We also added Figure 12 with power maps that explains the oscillatory behavior in Figure 13, and references to other works, including our studies.
- In section 5.2 the penultimate paragraph (line 453), what is the definition of ('its most linear section.')? what method is used to quantify it? least-square method? or other methods? Similarly, what is the definition of ('nonlinerity') in section 5.2 the last paragraph (line 469)?
Answer: We took the linear section limited by a range from -0.01 to +0.01 g that satisfies the linear function. We supplemented details in line 493. The nonlinearity was determined by the equation added in line 503.
- In Fig. 4, what's the advantage and disadvantage of the two types of MSEs, which may be the reason why these two types are put together to be compared.
Answer: Two types of MSE were integrated to investigate the dimensions of the proof mass. The mass selection was stipulated by the technology. More details (pros and cons) are given in lines 235-255.
- In page 7 the last paragraph (line 241), why (' the increase of its linear dimensions decreases its proof mass stiffness')? whether it is based on Eq. 3, the increase in 'a' decreases the 'h', thus leading ot the stiffness decreasing, when 'm' is a constant?
Answer. Here, we specifically meant the stiffness of the proof mass, because for Type 1 MSE it represents a plate with a thickness appreciably smaller that its dimension v (as stated above). This induces the bending of this plate under an inertial or gravitational acceleration load. Such bending also induces an inflection of the moving waveguide, which should be taken into account as well. We did not describe this issue and presented no calculations, because we selected Type 2 MSE where height h and dimension v are commeasurable. We have corrected the sentence to be more precise (line 251).
Reviewer 3 Report
Comments and Suggestions for Authors
In conclusion, in my opinion, this article may be of interest to the readers of the Sensors journal – especially to specialists in the particular field. Therefore, I recommend its publication after the indicated minor corrections are introduced:
1st remark. The article asserts that these systems are immune against electromagnetic noise. However, in the system proposed by the authors, optoelectronic elements are included, which are susceptible to external noise. Have any studies been conducted on the selection of elements aimed at reducing the impact of noise?
2nd remark. The article does not provide information on the accuracy of measurement. Have any studies been conducted to calibrate the accelerometer, for example with a reference system, signal, or accelerometer?
3rd remark. I would recommend that the conclusion include text that describes the areas of application for the accelerometer, according to the authors.
Author Response
Thank you for an objective peer-review of our article and the comments provided; this allowed us to improve the presentation of our research results.
1st remark. The article asserts that these systems are immune against electromagnetic noise. However, in the system proposed by the authors, optoelectronic elements are included, which are susceptible to external noise. Have any studies been conducted on the selection of elements aimed at reducing the impact of noise?
Answer. In our work, electromagnetic (EM) noise of the measuring transducer is meant. That is, the transducer can be used in locations with high EM noise. For instance, fiber optics can be used to transmit optical signal to and from the transducer. However, you are absolutely right that noise will affect the electronic part as in any other type of measuring transducer.
Other authors note the immunity of the optical transducer to EM noise, while omitting the issue under discussion. This is very reason why we also omitted it. In the present work, we used laboratory-grade power sources, optical radiation sources and optical power meters. In further studies, we will fabricate a transducer with a photodiode and laser on a single chip, and will certainly consider this issue when selecting electronic components.
2nd remark. The article does not provide information on the accuracy of measurement. Have any studies been conducted to calibrate the accelerometer, for example with a reference system, signal, or accelerometer?
Answer. We plan to fabricate a system with feedback and electronics, and bond input and output optic fibers to the optical outlets of the chip. Then, all the system will be suitable for installation on a rotating or vibrating platform to compare its performance with the reference system.
In this case, only a concept of the transducer itself was worked out. The power sources are stabilized. Currently, the optical fiber is placed without bonding to the optical outlets of the chip using micropositioners (Figure 18). This disallows adjusting the angle to the horizontal that can be used to set the gravity vector and make a comparison with the reference system. The noncontact input of optical radiation into the chip during the alteration of the angle of chip inclination to the horizontal will introduce extra error, which will negatively affect the experimental results. Optical losses of optical radiation input-output will differ every time the angle is changed, so it is impossible to predict their value in practice.
3rd remark. I would recommend that the conclusion include text that describes the areas of application for the accelerometer, according to the authors.
We have supplemented the conclusion with anticipated application sphere of the MOEMA under development (line 534).
Round 2
Reviewer 2 Report
Comments and Suggestions for Authors
All the comments and suggestions have been responsed reasonably. The authors have made efforts to add some discussions and analyses. The manuscript has been significantly improved.
Comments on the Quality of English LanguageThe quality of English has been significantly improved.